# Attitudes toward the Integration of Radiographers into the First-Line Interpretation of Imaging Using the Red Dot System

**DOI:** 10.3390/bioengineering10010071

**Published:** 2023-01-05

**Authors:** Ammar A. Oglat, Firas Fohely, Ali AL Masalmeh, Ismail AL Jbour, Laith AL Jaradat, Sema I. Athamnah

**Affiliations:** 1Department of Medical Imaging, Faculty of Applied Medical Sciences, The Hashemite University, Zarqa 13133, Jordan; 2Department of Medical Imaging, Faculty of Allied Medical Science, Palestine Ahliya University, Bethlehem 1041, Palestine; 3Biomedical Engineering Department, Faculty of Engineering, Jordan University of Science and Technology, Irbid 22110, Jordan

**Keywords:** red dot system, radiography reporting, clinical reporting radiographer, abnormality detection

## Abstract

The red dot system uses expertise in the identification of anomalies to assist radiologists in distinguishing radiological abnormalities and managing them before the radiologist report is sent. This is a small step on the road to greater role development for radiographers. This practice has existed for more than 20 years in the UK. Today, it is only the UK seeking to legislate radiographer reports. The aim of this paper is to put focus on this issue, determine whether radiographer reports are necessary, and explore whether there are any benefits that can be highlighted to encourage health authorities worldwide to allow radiographers to write clinical reports. Additionally, this study was conducted to evaluate the role of radiographers (non-radiologists) in medical image interpretation, using 95 samples that were collected randomly and a representative sample of radiographers and radiologists of both genders. The SPSS program was used for the statistical analysis of the samples and to scientifically explain the results. We found that radiologists have no objections to the participation of radiographers in diagnosis assistance, interpretation, and clinical reporting through the red dot system. Therefore, there was support for the future implementation of such a system in health care.

## 1. Introduction

Medical imaging is now an integral part of the whole continuum of health care; before, physicians did not have access to radiographs of the body of a patient [1,2,3,4,5]. However, Roentgen found a new kind of ray when testing with a Crookes cathode ray tube that he called X-rays. Roentgen’s discovery set in motion a series of medical diagnostics [6]. The revolution continues to this day [7,8,9,10,11]. Worldwide, radiographers are central to the diagnostic pathway and have expertise in radiographs. The radiographer, by nature, is the first healthcare practitioner to evaluate each image acquired. Furthermore, technicians (radiographers) are in a unique position to connect with and obtain expert impressions of the procedure directly from the clinician in a timely manner and thus have a significant influence on patient care [12,13,14,15,16,17,18,19,20]. 

The area of medicine that uses imaging examinations and procedures to diagnose pathologies and abnormalities as part of the treatment plan is called diagnostic radiology. Diagnostic radiology plays an important role in the diagnosis of disease or injury in any form of medical treatment. Radiation is also used to produce full anatomical images at stages that have been determined to be harmless. Diagnostic radiology, with its emphasis on diagnosis, represents a collaboration between the clinical and radiology teams. In such an environment, interpersonal experiences and communication breakdowns can lead to patient harm.

In the early 1980s, the red dot was introduced in the UK and was primarily used for trauma patients. The radiographer compares an image to others in this system and annotates or flags the image if they feel that there is an abnormality in order to convey this to the patient management clinician [21]. The emergence of the red dot system stems from the rapid growth of demand for diagnostic imaging services, which led to a widening gap between the demand for services and the ability to meet this demand, considering the small number of radiologists compared to the number of images, which led to the accumulation of many images without a diagnosis [22]. The lack of a “red mark” does not indicate that the radiographer did not detect an abnormality but may indicate that the radiographer was too busy studying the image, that they were not interested in the red dot system, or that they were not capable of making a decision on the normality of the image, etc. [22,23,24,25,26,27]. Some agencies have adopted a “traffic signal” red dot system to combat this confusion. A red dot means that the radiographer discovered an anomaly, an amber/yellow dot means that the radiographer is unable to assess image normality, a green dot means that the radiographer did not find an abnormality, and no dot means that the image was not checked by the radiographer for abnormalities [28]. An improved degree of coordination between the radiography team and the referencing team within a hospital setting is among the major benefits of this method. The red dot system ensured images flagged with a red received greater inspection and this minimized overall errors by the referring doctors [12,29,30].

This story begins with technologists, who reported findings in imaging applying the “red dot” and thus helping physicians (radiologists) with existing explanations of any disease. In World War II, there was a lack of radiologists to interpret the X-ray images, so Weber State University joined with the US Army, teaching large numbers of technologists (radiographers) to create radiology diagnoses for patients (soldiers). Currently, radiographers in the US can interpret radiographs if they pass legislation that legalizes radiographers interpreting radiographs, sonograms, or CT scans.

Radiographers have no legal diagnostic obligation to treat their patients at this time. In other words, radiographers are under no duty to inform any individual medical practitioner about an abnormal examination and, as such, they cannot be held responsible for withholding their opinions concerning the results of the examination [31,32,33]. Some researchers conducted a study to examine how trauma radiographs are recorded in a timely manner by radiologists. The study found a lack of timely radiograph reports and increased mortality rates and deaths in the hospitals examined [34]. Therefore, the red dot system needs to be introduced to assist emergency doctors to make quick decisions and a more detailed diagnosis despite the lack of a radiologist [34]. 

Radiographers’ professional scope and national curricula vary in Europe. In some countries, radiography includes both diagnostic imaging and radiotherapy at the Bachelor’s degree level. Even though the professional scope varies, a general aspect of this practice is its combination of operationally complex technology with human communication and patient care. Radiographers’ own professional experience of how these aspects blend together in practice has not been investigated. Increased knowledge of this topic would be useful for professionalization and educational purposes. Previous studies can be categorized into three major areas. 

Furthermore, these studies can be used to assess the willingness of radiographers to apply and adhere to this system, as well as to assess the acceptability of this system among radiologists.

The research hypotheses are detailed below:

**H1.** 
*There are no significant differences at the 0.05 level of significance in the percentages of respondents’ attitudes toward the role of the radiographer in radiological reports due to gender, age, job name, job place, years of experience, and qualification.*


**H2.** 
*There are no significant differences at the 0.05 level of significance in the percentages of respondents’ attitudes about to what extent the participation of radiologists in diagnosing can be useful and whether they have sufficient experience and education to participate in this system due to gender, age, job name, job place, years of experience, and qualification.*


**H3.** 
*There are no significant differences at the 0.05 level of significance in the percentages of respondents’ attitudes toward the effects and results of adding the red dot system due to gender, age, job name, job place, years of experience, and qualification.*


## 2. Methods

### 2.1. Study Design 

The current study was based on the cross-sectional method and was conducted for several reasons, such as: to highlight and evaluate the role of the radiographer in diagnosis; to assess the willingness of the radiographer to apply and adhere to this system; and to evaluate the extent of acceptability of radiologists in the application of this system. The study took place in the South West Bank/Palestine, and it included the main three public hospitals and the other five private hospitals.

### 2.2. Sampling

A sample of 95 patients was studied and collected randomly. This sample consisted of 20 radiologists and 75 radiographers; 73 of them were males, and 22 were females with different working experiences. However, in his research, all other staff in the radiology department were excluded, except radiographers and radiologists.

Regarding the age variable, the percentage of the category (20–29) was 47.4%; the percentage of the category (30–39) was 43.2%; and the percentage of the category (40 or more) was 9.5%. Regarding the job name variable, the percentage of radiologists was 21.1%, and the percentage of radiographers was 78.9%. Regarding job places, the percentage of the category (hospital) was 66.3%, and the percentage of the category (radiology center) was 33.7%. Regarding the years of experience variable, the percentage of the category (up to 5 years) was 51.6%, the percentage of the category (6–10) was 32.6%, and the percentage of the category (more than 10 years) was 15.8%. Regarding the qualification variable, the percentage of the category (BA) was 86.3%, and the percentage of the category (Master or Above) was 13.7%. See Table 1.

### 2.3. Data Collection 

The data collection in this study was based on a self-administered questionnaire. Ninety-five questionnaires were distributed randomly to the targeted samples.

The questionnaire consists of two sections. Section 1 includes some personal information about the person who will answer the questionnaire (sample study), including gender, age, job title, job place, years of experience, and qualification, whereas Section 2 was divided into three groups of questions to evaluate the role of radiographers in providing diagnostic assistance, assess the willingness of the radiographer to apply to this system, and evaluate the extent of acceptability of radiologists in the application of this system.

Group 1 was intended to clarify the role of the radiographers in radiological reports, whereas group two was intended to determine to what extent the participation of radiologists in diagnosing can be useful, and whether they have sufficient experience and education to participate in this system. Lastly, group three was designed to show the effects and results of adding the red dot system. The questionnaire was distributed to the targeted sample on the same day, under the supervision of the researcher.

### 2.4. Data Analysis

The researchers coded the data collected through the questionnaires and performed the needed data manipulation and statistical analysis using a computer statistical package for social science (SPSS) to screen and analyze the collected data. Each item was coded and given 1 point for a “Yes” or “Agree” answer and 2 points for a “No” or “Disagree” answer. After that, all the scores of the items were summed up, and the total was divided by the number of the items, giving a mean score for each part. These scores were converted again by scoring all values (1–1.49) into 1 point and (1.5–2) into 2 points.

The statistical methods used in the analysis of the research include frequencies and percentages to describe personal and demographic characteristics. Chi-square tests were used for testing the hypothesis of differences in respondents’ answers according to personal and demographic characteristics, as well as the Alpha (Cronbach) scales.

## 3. Results

The results of Table 2 show that radiologists and radiographers have a high agreement percentage toward the role of radiographers in radiological reports (82.1%).

The respondents agreed that the radiographer has the ability to communicate with the patient, which helps in the diagnosis by 88.3%; they also agreed that the professional observations of the radiographer can benefit directly from the diagnosis and thus have a great impact on patient care by 84.2%; they also agreed that the radiographer has the ability to improve diagnosis and image quality by 76.8%; and these activities should increase their job satisfaction and further develop their professional standing by 75.3%.

On the other hand, the respondents agreed that the participation of the radiographer in the diagnostic process increases the work pressure on him by only 54.7%, and the radiographer reported reducing patient waiting time by 44.7%, and they agreed with the radiographer to put a red dot on all types of imaging modalities by 44.4%.

Table 2 shows to what extent radiologists’ involvement in diagnosing can be useful and whether they have the necessary experience and education to participate in this system.

The results in Table 3 represent that the radiologists and radiographers have a high agreement percentage about to what extent the participation of radiologists in diagnosing is useful and whether they have sufficient experience and education to participate in this system (77.9%).

The following Table 3 displays the frequencies and percentages of radiologists and radiographers toward the effects and results of adding the red dot system.

The results of the table above (Table 3) show that the radiologists and radiographers have a high agreement percentage toward the effects and results of adding the red dot system and whether they have sufficient experience and education to participate in this system (70.5%).

The respondents agreed that educational materials and special training should be added to improve the radiographer’s ability to report by 87.2%, they agreed that the radiographer’s involvement in the diagnosis saves the radiologist’s time and makes them available to report other images by 71.6%, they agreed that radiographers can report X-ray examinations that reduce diagnostic errors, especially in the emergency department, by 71.6%, and they agreed that imaging diagnosis should be an important factor in the university’s evaluation of a radiographer student by 70.2%.

Furthermore, the respondents in Table 4 agreed that the participation of a radiographer in diagnostics helps bridge the gap in diagnostic imaging services by 62.1% and agreed that expanding the role of a radiographer in a specific set of diagnostic radiological reporting tasks could help meet the demand and relieve some of the pressure on radiologists by 60.6%. However, they agreed that the participation of the radiographer in the diagnosis compensates for the shortage of radiologists only by 42.1%. Moreover, the respondents agreed that cooperation between a radiologist and a radiographer in preparing reports leads to improvement in the quality and results of the diagnosis by 78.9%; they agreed that placing a red dot (badge or mark) of the radiographer on the image helps in diagnosing and reduces the percentage of errors in the diagnosis by 78.7%, and they agreed that a red dot image will receive more scrutiny by referring doctors and reduce diagnostic errors overall by 76.8%; they agreed that this system provides a higher level of communication between the radiography team and the referral team within the hospital environment by 75.8%; and they think that the red dot system allows the radiographer contribute to the diagnosis without fear of legal danger in particular by 70.5%.

Additionally, in Table 4, the respondents agreed that new imaging technology solutions in radiology should promote the development of radiographic skills and their use in a red point system environment by 69.5%, but the respondents agreed that X-ray departments are often unable to meet the goals set to provide general practitioner satisfaction with a rapid reporting service only by 52.6%, and only 37.9% of the respondents think that the health care system in Palestine is able to understand and adapt to the red point system.

There are no significant differences at the 0.05 level of significance in the percentages of respondents’ attitudes toward the role of the radiographer in radiological reports due to gender, age, job name, job place, years of experience, and qualification. To test this hypothesis, cross-tabulations (frequency and percentages) with chi-square test analysis were used to test differences in the percentages of respondents’ attitudes toward the role of the radiologist in radiological reports due to gender, age, job name, job place, years of experience, and qualification, and the results are shown in the following table:

The results in the table below (Table 5) show that there are significant differences at the 0.05 level of significance in percentages of respondents’ attitudes toward the role of the radiologist in radiological reports only due to qualification, as the *p*-value of the chi-square test (0.004) is less than 0.05. The results show that the BA-qualified respondents have a percentage of agreement of 86.6%, which is significantly higher than the percentage of agreement of 53.8% for Master or Above-qualified respondents.

On the other hand, the results in the table above show that there are no significant differences at the 0.05 level of significance in the percentages of respondent’s attitudes toward the role of the radiologist in radiological reports due to gender, age, job name, job place, and years of experience as the *p*-values of the Chi-Square test corresponding to these variables are higher than 0.05.

The final conclusion is that we reject hypothesis H1 regarding qualification and accept hypothesis H1 regarding gender, age, job name, job place, and years of experience. 

There are no significant differences at the 0.05 level of significance in the percentages of respondents’ attitudes about to what extent the participation of radiologists in diagnosing is useful, and whether they have sufficient experience and education to participate in this system due to gender, age, job name, job place, years of experience, and qualification. To test this hypothesis, cross-tabulations (frequency and percentages) with chi-square test analysis were used to test differences in the percentages of respondents’ attitudes about to what extent the participation of radiologists in diagnosing can be useful, and whether they have sufficient experience and education to participate in this system due to gender, age, job name, job place, years of experience, and qualification, and the results are shown in the following table:

The results in the table below (Table 6) show that there are no significant differences at the 0.05 level of significance in percentages of respondents’ attitudes about to what extent the participation of radiologists in diagnosing can be useful, and whether they have sufficient experience and education to participate in this system only due to gender, age, job name, job place, years of experience, and qualification as the *p*-values of the chi-square test corresponding to these variables are higher than 0.05. Therefore, the conclusion is that we accept hypothesis H2 regarding gender, age, job name, job place, years of experience, and qualification.

Here, are no significant differences at the 0.05 level of significance in the percentages of respondents’ attitudes toward the effects and results of adding the red dot system due to gender, age, job name, job place, years of experience, and qualification. To test this hypothesis, cross-tabulations (frequencies and percentages) with chi-square test analysis were used to test differences in the percentages of respondent’s attitudes toward the effects and results of adding the red dot system due to gender, age, job name, job place, years of experience, and qualification, and the results are shown in the following table.

The results in the table below (Table 7) show that there are significant differences at the 0.05 level of significance in the percentages of respondents’ attitudes toward the effects and results of adding the red dot system only due to years of experience and qualification, as the *p*-values of the chi-square test (0.040 and 0.006) are less than 0.05. The results show that the respondents who have (6–10) years of experience have a percentage of agreement of 80.6%, and the respondents who have (more than 10) years of experience have a percentage of agreement of 86.7%, and both of these percentages are significantly higher than the percentage of agreement of 59.2% for the respondents who have (up to 5) years of experience. The results also show that BA-qualified respondents have a percentage of agreement of 75.6%, which is significantly higher than the percentage of agreement of 38.5% for Master or Above-qualified respondents. On the other hand, the table shows that there are no significant differences at the 0.05 level of significance in percentages of respondents’ attitudes toward the effects and results of adding the red dot system due to gender, age, job name, and job place as the *p*-values of the chi-square test corresponding to these variables are higher than 0.05. The final conclusion is that we reject hypothesis H3 regarding the years of experience and qualification and accept hypothesis H3 regarding gender, age, job title, and job place.

## 4. Discussion 

This study was conducted to evaluate the role of the radiographer in providing diagnostic assistance. It assesses the extent of acceptance and readiness of both radiologists and radiographers to adhere to and apply this system in Palestine to assist radiologists in the diagnostic process, thus reducing the workload and stress on them.

The first hypothesis indicates that there are no significant differences in the role of the radiographer in radiological reports due to gender, age, job name, job place, and years of experience. This hypothesis aimed to assess the respondents’ opinions about the participation and role of the radiographer in radiological reports. However, the first hypothesis regarding the qualification variable was rejected as the *p*-value was less than 0.05. According to the radiologist, there appeared to be an objection in terms of scientific qualification, so the radiographer must have a Master’s degree or higher in order to be able to participate in the radiological reports.

However, the second hypothesis indicates that there are no significant differences in how much radiologists can help with diagnosis and whether they have enough experience and education to be involved in this system based on gender, age, job title, job location, years of experience, and qualification. This hypothesis measured the efficiency of the radiographer’s participation in the diagnostic process. Given that the *p*-values corresponding to these variables are higher than 0.05, this means that all respondents want to participate with a radiographer with assistance in the diagnostic process.

Finally, the third hypothesis indicates there are no statistically significant differences between the effects and results of adding the red point system according to gender, age, job name, and workplace, but the third hypothesis related to years of experience and qualifications variable was rejected. As the *p*-values are less than 0.05, it has been found that there is a comment on the part of the radiologist that a radiographer with less than 5 years of experience is not helpful and preferably has experience of 10 years or above to be of benefit in this system, and there was also an objection on the part of the radiologist for the scientific qualification of the radiographer and that it is better to have a Master’s degree or above, and that a radiographer with a BA degree will not be of any benefit for their participation.

During this study, it was found that there were no obstacles on the part of the radiologist and radiographer in Palestine to the participation of the radiographer in medical images and the clinical report, and this result encourages the application of the red dot system in the Palestinian health system.

## 5. Crucial Point about Radiographic Reporting

In the United Kingdom, when they asked radiographers to build and add diagnostic imaging programs, the radiographers initially made some mistakes in their reports; however, the government did not close the programs down or avoid radiographers reporting. The government allows radiographers to learn more and obtain more experience, and with time, radiographers had to adjust to the new work roles and began to learn how to produce acceptable diagnoses. If any country starts a radiographers’ reporting system, they will be required to give the radiographers some time and supervision until they succeed.

## 6. Proposed Programs and Advanced Practice

While several countries sustain from radiologists’ reduction, it is crucial to have more modern programs and practices for radiographers. From radiographers’ graduation from high school, they view various radiologic scans in their undergraduate and postgraduate degrees and throughout their clinical training. Furthermore, they are more invested than any medical doctor who just finished any MBBS or MD degree.

## 7. Conclusions

The red dot was introduced in the UK and was primarily used by trauma patients, which led to a widening gap between the demand for services and the ability to meet this demand, as well as the small number of radiologists compared to the number of images, which led to the accumulation of many images without a diagnosis, which increases the pressure of work on them. This is why we suggest creating a protocol through which we help radiographers with diagnosis by placing markers on the abnormal part in an image, which leads to reducing diagnostic errors. Therefore, as a conclusion, there is no objection by the radiologist to the participation of the radiographer in proving diagnosis assistance, interpretation, and clinical reporting through the red dot system. Therefore, there was support for the future implementation of such a system in the healthcare system.

## Figures and Tables

**Table 1 bioengineering-10-00071-t001:** Frequencies and percentages of the personal and demographic variables of the study sample.

Variable	Category	Frequency	Percent
Gender	Male	73	76.8%
	Female	22	23.2%
	Total	95	100.0%
Age	20–29	45	47.4%
	30–39	41	43.2%
	40 or more	9	9.5%
	Total	95	100.0%
Job Name	Radiologist	20	21.1%
	Radiographer	75	78.9%
	Total	95	100.0%
Job Place	Hospital	63	66.3%
	Radiography center	32	33.7%
	Total	95	100.0%
Years of Experience	up to 5 years	49	51.6%
	6–10	31	32.6%
	more than 10 years	15	15.8%
	Total	95	100.0%
Qualification	BA	82	86.3%
	Master or Above	13	13.7%
	Total	95	100.0%

**Table 2 bioengineering-10-00071-t002:** Frequencies and percentages of radiologists and radiographers toward the role of the radiographers in radiological reports (n = 95).

Item	Yes		No	
	N	%	N	%
The radiographer has the ability to communicate with the patient, which helps in the diagnosis?	83	88.3%	11	11.7%
Can the professional observations of the radiographer benefit directly from the diagnosis, and thus have a great impact on patient care?	80	84.2%	15	15.8%
Can radiographer reports reduce patient waiting time?	42	44.7%	52	55.3%
Should these activities increase their job satisfaction and further develop their professional standing?	70	75.3%	23	24.7%
Do radiographers have the ability to improve diagnosis and image quality?	73	76.8%	22	23.2%
Does the participation of the radiographer in the diagnostic process increase the work pressure on him?	52	54.7%	43	45.3%
Do you agree to put a radiographer red dot on all types of imaging modalities or on a specific type?	36	44.4%	45	55.6%
Total	78	82.1%	17	17.9%

**Table 3 bioengineering-10-00071-t003:** Frequencies and percentages of radiologists and radiographers about to what extent the participation of radiologists in diagnosing can be useful, and whether they have sufficient experience and education to participate in this system (n = 95).

Item	Yes		No	
	N	%	N	%
The participation of a radiographer in diagnostics helps bridge the gap in diagnostic imaging services?	59	62.1%	36	37.9%
Could expanding the role of radiographers in a specific set of diagnostic radiological reporting tasks help meet the demand, and relieve some of the pressure on radiologists?	57	60.6%	37	39.4%
The participation of the radiographer in the diagnosis compensates for the shortage of radiologists?	40	42.1%	55	57.9%
The radiographer’s involvement in the diagnosis saves the radiologist’s time and makes them available to report to other images?	68	71.6%	27	28.4%
Can radiographers report x-ray examinations that reduce diagnostic errors, especially in the emergency department?	68	71.6%	27	28.4%
Should imaging diagnosis be an important factor in the university’s evaluation of a radiographer student?	66	70.2%	28	29.8%
Should educational materials and special training be added to improve the radiographer’s ability to report?	68	87.2%	10	12.8%
Total	74	77.9%	21	22.1%

**Table 4 bioengineering-10-00071-t004:** Frequencies and percentages of radiologists and radiographers toward the effects and results of adding the red dot system (n = 95).

Item	Yes		No	
	N	%	N	%
By placing a red dot (badge or mark) on the radiographer on the image, it helps in diagnosing and reduces the percentage of errors in the diagnosis.	74	78.7%	20	21.3%
Cooperation between a radiologist and a radiographer in preparing reports that leads to improvement in the quality and results of the diagnosis?	75	78.9%	20	21.1%
X-ray departments are often unable to meet the goals set to provide general practitioner satisfaction with a rapid reporting service?	50	52.6%	45	47.4%
Should new imaging technology solutions in radiology promote the development of radiographic skills and their use in a red point system environment?	66	69.5%	29	30.5%
The red dot system allows the radiographer to contribute to the diagnosis without fear of legal danger in particular?	67	70.5%	28	29.5%
Do you think that the health care system in Palestine is able to understand and adapt to the red point system?	36	37.9%	59	62.1%
Will a red dot image receive more scrutiny by referring doctors and reduce diagnostic errors overall?	73	76.8%	22	23.2%
This system provides a higher level of communication between the radiography team and the referral team within the hospital environment?	72	75.8%	23	24.2%
Total	67	70.5%	28	29.5%

**Table 5 bioengineering-10-00071-t005:** The role of the radiologist in radiological reports. due to gender, age, job name, job place, years of experience, and qualification.

Demographic Factors		Total Respondents’ Attitudes toward the Role of the Radiographer in Radiological Reports			
		Yes	No	Chi-Square	*p*-Value
		N(%)	N(%)		
Gender	Male	60 (82.2%)	13 (17.8%)	0.002	0.968
	Female	18 (81.8%)	4 (18.2%)		
Age	20–29	36 (80%)	9 (20%)	0.436	0.804
	30–39	34 (82.9%)	7 (17.1%)		
	40 or more	8 (88.9%)	1 (11.1%)		
Job Name	Radiologist	17 (85%)	3 (15%)	0.144	0.704
	Radiographer	61 (81.3%)	14 (18.7%)		
Job Place	Hospital	51 (81%)	12 (19%)	0.169	0.681
	Radiography center	27 (84.4%)	5 (15.6%)		
Years of Experience	up to 5 years	36 (73.5%)	13 (26.5%)	5.199	0.074
	6–10	28 (90.3%)	3 (9.7%)		
	more than 10 years	14 (93.3%)	1 (6.7%)		
Qualification	BA	71 (86.6%)	11 (13.4%)	8.186	0.004
	Master or Above	7 (53.8%)	6 (46.2%)		

**Table 6 bioengineering-10-00071-t006:** Respondents’ attitudes about to what extent the participation of radiologists in diagnosing can be useful, and whether they have sufficient experience and education to participate in this system due to gender, age, job name, job place, years of experience, and qualification.

Demographic Factors		Total Respondents’ Attitudes about to What Extent Can the Participation of Radiologists in Diagnosing Be Useful, and Does He Have Sufficient Experience and Education to Participate in This System			
		Yes	No	Chi-Square	*p*-Value
		N(%)	N(%)		
Gender	Male	57 (78.1%)	16 (21.9%)	0.006	0.936
	Female	17 (77.3%)	5 (22.7%)		
Age	20–29	36 (80%)	9 (20%)	0.240	0.887
	30–39	31 (75.6%)	10 (24.4%)		
	40 or more	7 (77.8%)	2 (22.2%)		
Job Name	Radiologist	16 (80%)	4 (20%)	0.065	0.798
	Radiographer	58 (77.3%)	17 (22.7%)		
Job Place	Hospital	47 (74.6%)	16 (25.4%)	1.177	0.278
	Radiography center	27 (84.4%)	5 (15.6%)		
Years of Experience	up to 5 years	37 (75.5%)	12 (24.5%)	0.836	0.658
	6–10	24 (77.4%)	7 (22.6%)		
	more than 10 years	13 (86.7%)	2 (13.3%)		
Qualification	BA	66 (80.5%)	16 (19.5%)	2.340	0.126
	Master or Above	8 (61.5%)	5 (38.5%)		

**Table 7 bioengineering-10-00071-t007:** Effects and results of adding the red dot system.

Demographic Factors		Total Respondents Attitudes toward the Effects and Results of Adding the Red Dot System			
		Yes	No	Chi-Square	*p*-Value
		N(%)	N(%)		
Gender	Male	50 (68.5%)	23 (31.5%)	0.627	0.429
	Female	17 (77.3%)	5 (22.7%)		
Age	20–29	29 (64.4%)	16 (35.6%)	2.399	0.301
	30–39	30 (73.2%)	11 (26.8%)		
	40 or more	8 (88.9%)	1 (11.1%)		
Job Name	Radiologist	15 (75%)	5 (25%)	0.244	0.621
	Radiographer	52 (69.3%)	23 (30.7%)		
Job Place	Hospital	47 (74.6%)	16 (25.4%)	1.495	0.221
	Radiography center	20 (62.5%)	12 (37.5%)		
Years of Experience	up to 5 years	29 (59.2%)	20 (40.8%)	6.440	0.040
	6–10	25 (80.6%)	6 (19.4%)		
	more than 10 years	13 (86.7%)	2 (13.3%)		
Qualification	BA	62 (75.6%)	20 (24.4%)	7.449	0.006
	Master or Above	5 (38.5%)	8 (61.5%)		

## Data Availability

Not applicable.

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
