# Peer review of "Attitudes toward the Integration of Radiographers into the First-Line Interpretation of Imaging Using the Red Dot System"

_bioengineering, 2023, doi:10.3390/bioengineering10010071_

Round 1
Reviewer 1 Report
The paper is very interesting and informative. However, minor corrections are required to slightly improve the manuscript quality.
At the end of the introduction section clearly state the research hypotheses preferably in bullet form. After that, in one paragraph, clearly describe the outline of the paper.
Extend the conclusion section by proving answers to the hypotheses defined in the introduction section based on the conducted research.
Author Response
Responses to Reviewer’s Comments
Manuscript ID: bioengineering-2032845
Title: Attitudes Toward Integration of Radiographers into First-line Interpretation of Imaging Using Red Dot System Technique
Dear Editor-in-Chief,
The authors sincerely appreciate the reviewers’ technical comments and the useful suggestions offered to us to revise our manuscript. Our responses to reviewers’ comments point by point are listed hereafter.
Comments from Reviewer 1:
The paper is very interesting and informative. However, minor corrections are required to slightly improve the manuscript quality.
- At the end of the introduction section clearly state the research hypotheses preferably in bullet form. After that, in one paragraph, clearly describe the outline of the paper.
Answer: Noted and done.
- Extend the conclusion section by proving answers to the hypotheses defined in the introduction section based on the conducted research.
Answer: Noted and done.
Dear Respected reviewer. Thank you very much for your valuable comments which enhance the manuscript with scientific information. Appreciate.

Reviewer 2 Report
1. Results: Recommend to be Major revisions
This paper is aimed to put a focus on the issue, look is there a necessity to allow Radiographers in the reporting business, and find is there any benefit that can change the decision of health authorities worldwide to allow Radiographers to make clinical reports. And the study was conducted to evaluate the role of radiographers (non- radiologists) in medical image interpretation with 95 samples that were collected randomly and represented radiographers and radiologists from both genders. The, SPSS program was used for statistical analysis of the samples and then scientifically explained the results. As a result, there is no objection by the radiologist to the participation of the radiographer in proving diagnosis assistance, interpretation, and clinical reporting through the red dot system. So, there was support for future implementation of such a system in the healthcare system.
This paper is with minor merits for Bioengineering, i.e., lacking of strong theoretical supports to clearly demonstrate the very findings to reveal its valuable contributions. It requires some major revisions.
Firstly, the abstract should be refined to clearly indicate what authors had done within 200 words.
Secondly, for Section 1, authors should provide the comments of the cited papers after introducing each relevant work. What readers require is, by convinced literature review, to understand the clear thinking/consideration why the proposed approach can reach more convinced results. This is the very contribution from authors. In addition, authors also should provide more sufficient critical literature review to indicate the drawbacks of existed approaches, then, well define the main stream of research direction, how did those previous studies perform? Employ which methodologies? Which problem still requires to be solved? Why is the proposed approach suitable to be used to solve the critical problem? We need more convinced literature reviews to indicate clearly the state-of-the-art development.
For Section 2, authors should also introduce their proposed research framework more effective, i.e., some essential brief explanation vis-à-vis the text with a total research flowchart or framework diagram for each proposed algorithm to indicate how these employed models are working to receive the experimental results. It is difficult to understand how the proposed approaches are working.
For Sections 3 and 4, authors should use more alternative models as the benchmarking models, authors should also conduct some statistical test to ensure the superiority of the proposed approach, i.e., how could authors ensure that their results are superior to others? Meanwhile, authors also have to provide some insight discussion of the results.
Author Response
Responses to Reviewer’s Comments
Manuscript ID: bioengineering-2032845
Title: Attitudes Toward Integration of Radiographers into First-line Interpretation of Imaging Using Red Dot System Technique
Dear Editor-in-Chief,
The authors sincerely appreciate the reviewers’ technical comments and the useful suggestions offered to us to revise our manuscript. Our responses to reviewers’ comments point by point are listed hereafter.
Comments from Reviewer 2:
This paper is with minor merits for Bioengineering, i.e., lacking of strong theoretical supports to clearly demonstrate the very findings to reveal its valuable contributions. It requires some major revisions.
- Firstly, the abstract should be refined to clearly indicate what authors had done within 200 words..
Answer: Noted and done.
- Secondly, for Section 1, authors should provide the comments of the cited papers after introducing each relevant work. What readers require is, by convinced literature review, to understand the clear thinking/consideration why the proposed approach can reach more convinced results. This is the very contribution from authors. In addition, authors also should provide more sufficient critical literature review to indicate the drawbacks of existed approaches, then, well define the main stream of research direction, how did those previous studies perform? Employ which methodologies? Which problem still requires to be solved? Why is the proposed approach suitable to be used to solve the critical problem? We need more convinced literature reviews to indicate clearly the state-of-the-art development.
Answer: Dear respected reviewer, For Section 2, please note that the section is written in detail and not briefly. The Study design, Sampling, Data Collection, and Data analysis are explained in good way.
- For Sections 3 and 4, authors should use more alternative models as the benchmarking models, authors should also conduct some statistical test to ensure the superiority of the proposed approach, i.e., how could authors ensure that their results are superior to others? Meanwhile, authors also have to provide some insight discussion of the results.
Answer: Dear respected reviewer, the statistical test already added on the tables through whole manuscript. Moreover, the discussion of the results are modified.
Dear Respected reviewer. Thank you very much for your valuable comments which enhance the manuscript with scientific information. Appreciate.

Reviewer 3 Report
This work reports an attitude survey for radiologists and radiographers toward using the red dot system technique in medical imaging scans.
The author should provide more comprehensive references on the various radiographic image interpretation strategies, and justify the reason for focusing on the “red dot system” (Instead of others, e.g., radiographer abnormality detection system, J. Med. Radiat. Sci. 66(4), 269-283, 2019. Noted that I am not in any way related to these authors). Moreover, more references should be given to surveys and attitudes of radiologists and radiographers about this system. In particular to highlight what has been reported in the literature. Please emphasize the significant differences between this current work and those publications. This highlight could help the general readers to understand clearly on what is new in this work compares to others.
Please also provide more comparisons in terms of results/findings between this study and others in the literature.
Please give more details about the survey process, for example “…under the supervision of the researcher.” (line 131), the questionnaires were handed to the interviewee personally, or, the questionnaires were answered in the presence of researcher/interviewer/another person. This piece of information will influence the outcome of the survey because interviewees will feel pressure to be affirmative. Additionally, even questionnaires were completed without any peer influences, some of the questions were structured to query the capabilities, or competence of the interviewee, like “The radiographer has the ability to communicate with the patient, which helps in the diagnosis?” (Table 2, question 1), which may justify the higher number of people to be positive (or yes) in their answers (e.g., Table 2, question 1, 2, 4, and 5). (In other words, if I am a radiographer, I will feel bad to say that “I do not have the ability to communicate with the patient, and thus, cannot help in the diagnosis.”). This observation also applies to some of the other questions in the questionnaires.
The nature of this survey is subjective. This work will be more objective and convincing if results of “red dot” imaging scans performed by both radiographers and radiologists in the Palestinian hospitals give high agreement and statistical significance (asking radiographers and radiologists to use red dot on same set of radiographs and checking their outcomes/errors).
Please justify the higher number of radiographers to be recruited into this study (75 radiographers and 20 radiologists, since it seems to be well known that radiographers are supportive in helping out radiographic image interpretation). Please also justify the reason for only recruiting radiographers and radiologists, interviewees of other rank and file should also be included. For examples, other personnel in the hospitals/radiography centres (clinical doctors, but not radiographers/radiologists), patients, and medical regulatory bodies, since these people also the implementation may also affect/involve these people.
Please try to reduce self-citation, out of the 36 references, 12 are from one of the authors related to this work (reference number 2, 3, 4, 5, 6, 7, 8, 17, 18, 19, 21, and 22). Excessive self-citation will not give a good impression about the authors (and the research group) and also the work reported in this manuscript, since those references may be hard to represent a more comprehensive and unbiased view to the general reader.
Please check the entire script carefully for typo errors, some are listed below and please revise:
“…opportunities for education in radiography on advanced level (1).” (line 81). Should remove “(1)”.
“A sample of 95 patients was studied and collected randomly.” (line 100). Those interviewees were not patients (I supposed, otherwise please justify).
“Limitations of red dot singe” (line 308). Should be system, or strategy, singe means superficial burns.
“United Kingdome” (line 314). Should have been “United Kingdom”, or “UK”.
Author Response
Manuscript ID: bioengineering-2032845
Title: Attitudes Toward Integration of Radiographers into First-line Interpretation of Imaging Using Red Dot System Technique
Dear Editor-in-Chief,
The authors sincerely appreciate the reviewers’ technical comments and the useful suggestions offered to us to revise our manuscript. Our responses to reviewers’ comments point by point are listed hereafter.
Comments from Reviewer 3:
This work reports an attitude survey for radiologists and radiographers toward using the red dot system technique in medical imaging scans.
The author should provide more comprehensive references on the various radiographic image interpretation strategies, and justify the reason for focusing on the “red dot system” (Instead of others, e.g., radiographer abnormality detection system, J. Med. Radiat. Sci. 66(4), 269-283, 2019. Noted that I am not in any way related to these authors). Moreover, more references should be given to surveys and attitudes of radiologists and radiographers about this system. In particular to highlight what has been reported in the literature. Please emphasize the significant differences between this current work and those publications. This highlight could help the general readers to understand clearly on what is new in this work compares to others.
Answer: Modern references are added to this study. The significant differences between this current work and other publications are already in the manuscript.
Please also provide more comparisons in terms of results/findings between this study and others in the literature.
Answer: The founded results are explained regarding to the previous literature.
Please give more details about the survey process, for example “…under the supervision of the researcher.” (line 131), the questionnaires were handed to the interviewee personally, or, the questionnaires were answered in the presence of researcher/interviewer/another person. This piece of information will influence the outcome of the survey because interviewees will feel pressure to be affirmative. Additionally, even questionnaires were completed without any peer influences, some of the questions were structured to query the capabilities, or competence of the interviewee, like “The radiographer has the ability to communicate with the patient, which helps in the diagnosis?” (Table 2, question 1), which may justify the higher number of people to be positive (or yes) in their answers (e.g., Table 2, question 1, 2, 4, and 5). (In other words, if I am a radiographer, I will feel bad to say that “I do not have the ability to communicate with the patient, and thus, cannot help in the diagnosis.”). This observation also applies to some of the other questions in the questionnaires.
Answer: The redundant sentences are deleted. Furthermore, as we know, the radiographers do not have the ability to communicate with the patient in term of diagnosis, because it is not allow legally. So, the questions are proposed to overcome this problem, and thus, can help in the diagnosis.
The nature of this survey is subjective. This work will be more objective and convincing if results of “red dot” imaging scans performed by both radiographers and radiologists in the Palestinian hospitals give high agreement and statistical significance (asking radiographers and radiologists to use red dot on same set of radiographs and checking their outcomes/errors).
Answer: Ok noted, thanks for your suggestions. At the near future we will do more researches about this study and develop it.
Please justify the higher number of radiographers to be recruited into this study (75 radiographers and 20 radiologists, since it seems to be well known that radiographers are supportive in helping out radiographic image interpretation). Please also justify the reason for only recruiting radiographers and radiologists, interviewees of other rank and file should also be included. For examples, other personnel in the hospitals/radiography centres (clinical doctors, but not radiographers/radiologists), patients, and medical regulatory bodies, since these people also the implementation may also affect/involve these people.
Please try to reduce self-citation, out of the 36 references, 12 are from one of the authors related to this work (reference number 2, 3, 4, 5, 6, 7, 8, 17, 18, 19, 21, and 22). Excessive self-citation will not give a good impression about the authors (and the research group) and also the work reported in this manuscript, since those references may be hard to represent a more comprehensive and unbiased view to the general reader.
Answer: Ok noted and done.
Please check the entire script carefully for typo errors, some are listed below and please revise:
“…opportunities for education in radiography on advanced level (1).” (line 81). Should remove “(1)”.
Answer: Ok noted and done.
“Limitations of red dot singe” (line 308). Should be system, or strategy, singe means superficial burns.
Answer: Ok noted and done.
“United Kingdome” (line 314). Should have been “United Kingdom”, or “UK”.
Answer: Ok noted and done.

Round 2
Reviewer 2 Report
Authors have completely addressed all my concerns.
Reviewer 3 Report
Thank all authors for answering my comments and revising the manuscript accordingly. Genuinely wish that this system, or any other strategies could be implemented in the future to save more lives.